# A Nanocomposite Based on Reduced Graphene and Gold Nanoparticles for Highly Sensitive Electrochemical Detection of *Pseudomonas aeruginosa* through Its Virulence Factors

**DOI:** 10.3390/ma12071180

**Published:** 2019-04-11

**Authors:** Islem Gandouzi, Mihaela Tertis, Andreea Cernat, Dalila Saidane-Mosbahi, Aranka Ilea, Cecilia Cristea

**Affiliations:** 1Analytical Chemistry Department, Faculty of Pharmacy, Iuliu Haţieganu University of Medicine and Pharmacy, 4 Louis Pasteur St., 400349 Cluj-Napoca, Romania; islemgandouzi@yahoo.fr (I.G.); mihaela.tertis@umfcluj.ro (M.T.); ilioaia.andreea@umfcluj.ro (A.C.); 2Laboratory of Analysis, Treatment and Valorization of the Pollutants of the Environment and Products, Faculty of Pharmacy, University of Monastir, Avicenne St., 5000 Monastir, Tunisia; Dalila.Saidane@fphm.rnu.tn; 3Department of Oral Rehabilitation, Oral Health and Dental Office Management, Faculty of Dentistry, Iuliu Haţieganu University of Medicine and Pharmacy, 400012 Cluj-Napoca, Romania; arankailea@yahoo.com

**Keywords:** *Pseudomonas aeruginosa*, pyoverdine, graphene, electrochemical sensor, virulence, factors

## Abstract

Pyoverdine is a fluorescent siderophore produced by *Pseudomonas aeruginosa* that can be considered as a detectable marker in nosocomial infections. The presence of pyoverdine in water can be directly linked to the presence of the *P. aeruginosa*, thus being a nontoxic and low-cost marker for the detection of biological contamination. A novel platform was developed and applied for the electrochemical selective and sensitive detection of pyoverdine, based on a graphene/graphite-modified screen-printed electrode (SPE) that was electrochemically reduced and decorated with gold nanoparticles (NPs). The optimized sensor presenting higher sensitivity towards pyoverdine was successfully applied for its detection in real samples (serum, saliva, and tap water), in the presence of various interfering species. The excellent analytical performances underline the premises for an early diagnosis kit of bacterial infections based on electrochemical sensors.

## 1. Introduction

Nosocomial infections, known as hospital-acquired/associated infections, are caused by various bacteria species, the most frequent being *Streptococcus* species (spp.), *Acinetobacter* spp., *Pseudomonas aeruginosa*, *Staphylococcus aureus*, *Bacillus cereus*, *Legionella*, *Proteus mirablis*, *Klebsiella pneumonia*, *Escherichia coli* and *Serratia marcescens*. Out of these, *P. aeruginosa*, *S. aureus*, and *E. coli* are the most common bacteria related to the highest rate of disease and complications. The repercussions are prolonged hospital stays and a major risk of serious health issues leading to death, both from medical and economic point of view [1].

A mandatory check-point in the development of rapid tests for identifying bacteria is the identification of virulence factors, which are the key to survival, replication, and development of a disease [2,3]. Being essential for bacterial growth and replication, iron became the target for highly efficient iron-acquisition systems known as siderophores [4]. These iron chelators are clinical biomarkers for specific bacteria, commercially available at low costs and with valuable perspectives for early diagnosis and therapy modulation.

The most common bacteria siderophores are pyoverdine (PyoV) and pyochelin for *P. aeruginosa*, aerobactin and enterobactin for *E. coli*, and staphyloferrin A and staphyloferrin B for *S. aureus* [4]. In the past few decades, the search for new and improved tools for the detection of virulence factors has led to the development of biochemical methods, genetic screens, and transcriptomic and genomic approaches, especially for the most common bacteria [5,6]. Mass spectrometries, nanoparticle (NP)-assisted microextraction approaches for bacterial profiling, have been employed to track down and identify these iron-scavenging molecules [7,8]. Moreover, proteomic methods were applied for full protein mapping as an important tool for the detection of biomedical biomarkers. The major drawbacks of these methods are represented by restrictive laboratory conditions, the need for highly trained personnel, and high analysis costs. The sampling and transport procedures could also be prone to a high degree of contamination, which could generate false positive/negative conclusions. A suitable alternative may be represented by the elaboration of electrochemical sensors for the detection and quantification in situ of the analytes. This approach associates the high selectivity, sensitivity, and rapidity related to electrochemical methods with low production costs and stability. Also, the possibility of miniaturization and decentralized analysis enables the sensors to be integrated in point-of-care (POC) devices [9,10]. The generation of 3D nanoarchitectures for the immobilization of biomolecules represents the first major goal in the fabrication of a biosensor. The platforms based on conductive polymers are widely employed due to their biocompatibility, increased electron transfer rates, and augmented active surface areas [11,12,13]. The supplementary tailoring with carbon-based nanomaterials or metallic nanoparticles (NPs) enhances their features, with outstanding results in the analytical performances of the (bio)sensors. The explanation is that the association between nanomaterials can create a summed effect towards the detection of the analyte, combining both the increase of the active surface area with new catalytic properties and increased selectivity [14,15]. In our previous studies, two platforms for the detection of PyoV, based on Au chemically modified graphene and polypyrrole-functionalized carboxylic groups with AuNPs were developed, both with close results for the detection of the target analyte [16,17]. In order to achieve miniaturization and integration in portable sensing devices, the protocol was readapted to overcome the observed difficulties, specifically the instability of the Au-modified graphene layers and the low conductivity of the polymeric film due to its passivation. Thus, the protocol was adapted to the electrochemical deposition of AuNPs on the graphene/graphite working electrode instead of using chemically Au-modified graphene. The upgrade in the development protocol was closely linked to the increase of the stability of the platform as well as its sensitivity. Moreover, the polymeric film was eliminated, simplifying and reducing the time of the elaboration. The outcome is represented by a tailored hybrid surface with the same selectivity, but an improved sensitivity and limit of detection (LOD) compared to the other reported approaches.

## 2. Materials and Methods

### 2.1. Reagents

The reagents were purchased from Sigma-Aldrich (St. Louis, MO, USA), Fluka Chemie GmbH (Buchs, Switzerland), Merck Chemicals (Darmstadt, Germany), Alfa Aesar (Karlsruhe, Germany), and used without any further treatments. PyoV from *Pseudomonas fluorescens* (P8374), pyocyanin (PyoC) (P0046), sodium chloride (746398), potassium ferrocyanide (K_4_[Fe(CN)_6_]) (P3289), potassium ferricyanide (K_3_[Fe(CN)_6_]) (31253), acetylsalicylic acid (ASA) (PHR1003), *L*-ascorbic acid (AA) (A92902), *β*-nicotinamide adenine dinucleotide phosphate-reduced tetrasodium salt (NADH) (10107735001), glucose (G) (G8270), and dopamine (DA) (H8502) were purchased from Sigma-Aldrich, while uric acid (UA) (51449), disodium hydrogen phosphate (71650), and sodium dihydrogen phosphate (71504) were purchased from Fluka Chemie GmbH. Sulphuric acid (100732), nitric acid (101799), and ethanol (K48380027 645) were purchased from Merck Chemicals, and hexachloroauric acid (12325) from Alfa Aesar.

The stock solutions were prepared using double-distilled water and ultrapure water (Milli-Q, Millipore; 18 MΩ·cm^−1^). The preparation of samples was performed depending on the desired concentrations, as follows: PyoV dilutions of 0.5, 0.75, 1, 5, 10, 25, 50, 75, and 100 µM were prepared in 20 mM saline phosphate buffer (PBS) pH 7.4 from a stock solution of PyoV–Fe complex from *P. fluorescens* of 860 μM prepared in ultrapure water (MilliQ), and stored at −20 °C. The concentrations of possible interfering compounds were established close to real values in order to evaluate experimental conditions similar to those of real matrices, as follows: G 7000 µM (pathological concentration), while AA, ASA, UA, NADH, and DA solutions were of 25 µM. In all cases, the concentration of PyoV was kept constant at 25 µM. The effect of PyoC, another *P. aeruginosa* metabolite, was also assessed in the presence of 25 µM PyoV at a concentration of 10 µM in 20 mM PBS pH 7.4. The real sample analyses were performed on Pierce normal human serum purchased from Thermo Scientific and tap water. The stimulated saliva was collected in plastic sterilized containers (Salivette device, Thermo Ficher Scientific, Waltham, MA, USA) from a healthy volunteer. Consent from the participant was obtained prior to the collection of samples, and all local guidelines (No. 94/08.03.2017) regarding the work with human saliva were respected. The biological samples were prepared following a simple pretreatment protocol that involved a 1:100 dilution with 20 mM PBS pH 7.4 and a filtration process through a 0.2-µm pore diameter filter (Phenex), with storage at −20 °C. Tap water samples were diluted 1:1 with 20 mM PBS pH 7.4 and stored at 4 °C. Prior to experiments, the samples were spiked with 25 µM PyoV and tested without accumulation time.

### 2.2. Sensor Elaboration and Electrochemical Characterization

The sensors were developed on graphene/graphite-based screen-printed electrodes (SPE) (DropSens, Asturias, Spain) with an integrated three-electrode cell based on a graphene/graphite working electrode with a 4-mm diameter, a silver reference, and a carbon counter electrode. The electrochemical experiments were recorded using an Autolab PGSTAT100 potentiostat (Metrohm, Eco Chemie Netherlands, Utrecht, Netherlands) with a module for electrochemical impedance spectroscopy (EIS) using Nova 1.10.4 software.

The elaboration consisted in a simple two-step protocol that required less than 10 min preparation time:(i)The electrochemical reduction of graphene was performed by sweeping the potential 10 times from +0.100 to −1.500 vs. Ag/V (pseudo Ag/AgCl) with a scan rate of 0.100 V·s^−1^ in a 30-µL drop of 20 mM PBS pH 7.4;(ii)The electrochemical decoration of reduced graphene with AuNPs was done by cycling 10 times the potential from −0.2 to +1.2 vs. Ag/V (pseudo Ag/AgCl) with a scan rate of 0.100 V·s^−1^ in a 50-µL drop of 2 mM HAuCl_4_ prepared in 0.5 M H_2_SO_4_.

The electrochemical characterization of the sensor was tested step by step, after each modification level, in 10 mM [Fe(CN)_6_]^3−/4−^ redox shuttle in 20 mM PBS pH 7.4 solution by cyclic voltammetry (CV) and EIS. The CV experiments were recorded by sweeping the potential from −0.5 to +0.8 vs. Ag/V (pseudo Ag/AgCl), at a scan rate of 0.1 V·s^−1^. The EIS analyses were carried on a frequency window of 10 mHz to 100 KHz at open circuit potential (OCP). The electrochemical tests were performed in a 50-µL drop of redox mediator. The sensitivity of the sensor towards PyoV was assessed by differential pulse voltammetry (DPV). The oxidation potential of PyoV was recorded by sweeping the potential from −0.3 to +0.8 vs. Ag/V (pseudo Ag/AgCl) with a scan rate of 0.02 V·s^−1^ in a 25-µL drop of stock solution. When evaluating the interference effect of PyoC, the potential range was cathodically shifted from −0.5 to +0.8 vs. Ag/V (pseudo Ag/AgCl), while keeping the other parameters constant.

### 2.3. Morphological Characterization of the Sensor

The topography of the sensor was evaluated by scanning electron microscopy (SEM). The SEM images for the surface characterization were registered on SU8230 SEM (Hitachi, Tokyo, Japan) at an accelerating voltage of 30 kV, 10 mA extraction current, and a working distance of 11 mm from multiple points on the surface.

## 3. Results

### 3.1. Analytical Principle of the Electrochemical Sensor for PyoV Detection

The principle of the electrochemical PyoV sensor based on reduced graphene and AuNPs was presented in Scheme 1.

Firstly, the graphene/graphite-based electrode was electrochemically reduced in the presence of 20 mM PBS pH 7.4 by cyclic voltammetry (CV) (Figure 1A). As can be observed in Figure 1A, the signal decreases proportionally with the number of cycles.

After washing the electrode with ultrapure water, the AuNPs were electrochemically generated in the presence of HAuCl_4_ by CV (Figure 1B). The electrochemical deposition of AuNPs was confirmed by the increase of both anodic and cathodic peaks corresponding to the oxidation/reduction of Au on the graphene/graphite surface, as can be seen in Figure 1B.

After the complete modification of the graphene/graphite working surface, an oxidation peak was observed in the presence of PyoV at about +0.325 vs. Ag/V (pseudo Ag/AgCl).

The oxidation signal registered by using an optimized DPV procedure was proportional with the concentration of PyoV, the analyte being detected with a higher sensitivity than similar electrochemical sensors [16,17].

### 3.2. Electrochemical Characterization of the PyoV Sensor Using CV, DPV, and EIS

Electrochemical techniques such as CV, DPV, and EIS were used for the electrochemical characterization after each modification step during sensor elaboration. Both CV and EIS studies were performed in 10 mM [Fe(CN)_6_]^3−/4−^ (1:1) in 20 mM PBS pH 7.4. CV was used to track the changes of electrochemical behavior at different stages of sensor elaboration, and is presented in Figure 2A. A reversible redox peak appeared in all tested situations due to the equivalent amount of [Fe(CN)_6_]^4−^ and [Fe(CN)_6_]^3−^ ions in the solution. As presented in Figure 2A (black curve (a)), the CV for the bare graphene/graphite electrode registered the anodic peak at +0.168 vs. Ag/V (pseudo Ag/AgCl), while the cathodic peak appeared at −0.036 vs. Ag/V (pseudo Ag/AgCl), resulting in peak-to-peak separation of 0.204 V. After the electrochemical reduction of the graphene layer (Figure 2A (red curve (b)), a slight increase in oxidation/reduction peak currents, paired with an anodic shift was observed, maintaining the same peak separation of 0.200 V. This may be attributed to the removal of any impurity that could block the access of the redox probe to the surface. From Figure 2A (green curve (c)), it was observed that oxidation/reduction peak currents were consistently larger after the electrochemical decoration of the reduced graphene entities with AuNPs, a registered increase in peak current of 55%. 

The EIS plots represented in Figure 2B (red curve (b)) showed that the reduction of graphene did not influence the parameters in Nyquist representation of impedance compared with the bare graphene/graphite SPE (black curve (a)). Thus, similar values were obtained for the R_ct_ using the software facilities in both cases (23.87 Ω compared with 24.32 Ω). After the deposition of AuNPs, an important decrease of R_ct_ to 11.09 Ω was registered (green curve (c)) due to the large electron transfer speed and better electrical conductivity of AuNPs compared with that of the bare electrode. Furthermore, the Nyquist plot changed, confirming the modification of the electrochemical transformation mechanism at the surface of the working electrode after the completed protocol.

### 3.3. Amplification Performance of the Electrochemical PyoV Sensor

In order to verify the amplification performance of the electrochemical PyoV sensor, DPV tests were recorded step by step throughout the experimental protocol with and without the addition of the analyte (Figure 3). There was a remarkable current response of 14.67 µA at +0.325 vs. Ag/V (pseudo Ag/AgCl) on the optimized sensor in the presence of 100 µM PyoV (Figure 3, blue curve (d)), compared with a mild current response of only 5.04 μA at +0.350 vs. Ag/V (pseudo Ag/AgCl) on the bare graphene/graphite-based SPE (Figure 3, red curve (b)), and of 8.16 µA at +0.340 vs. Ag/V (pseudo Ag/AgCl) on the graphene/graphite-based SPE after the electrochemical reduction of graphene (Figure 3, green curve (c)). The control DPV (black curve (a)) registered in the absence of PyoV did not show any oxidation signal in the optimized potential range. Thus, the reduced graphene and AuNPs could co-mediate the electrochemical oxidation of PyoV amplifying the signal in DPV.

### 3.4. SEM Characterization of PyoV Sensor

Figure 4 shows the SEM images of the stepwise modification process of the PyoV sensor. Figure 4A,B reveals the SEM images of the bare SPE based on the graphene/graphite working surface before and after the decoration with AuNPs. On the other hand, SEM images were obtained on reduced graphene/graphite before and after the electrochemical deposition of AuNPs were presented, as shown in Figure 4C,D. It was observed that the surface of the electrode was homogenous, covered with metallic NPs after the electrochemical reduction of graphene, which confirms the importance of this step in the experimental protocol. As can be also seen, the coverage with AuNPs was higher after the conditioning step, furthermore confirming the importance of the conditioning protocol regarding the density in catalytic metallic NPs.

### 3.5. Analytical Performance of the PyoV Sensor

Under optimized experimental conditions, the PyoV electrochemical sensor was used to quantify different concentrations of the analyte using an optimized DPV procedure. The variation of the DPV current response with the concentration of PyoV is presented in Figure 5A. No oxidation signal was observed without the addition of PyoV in the tested solution (black curve), while noticeable oxidation signals appeared with the addition of PyoV starting from 0.5 µM. Moreover, the current response increased linearly with the increase of PyoV concentration from 0.5 to 100 μM (Figure 5B). The regression equation was fitted as I (µA) = 0.139[PyoV] (µM)–0.357 (R^2^ = 0.993; RSD = 5.36% (Relative standard deviation (RSD) was calculated as average of RSD for each concentration in the calibration curve)). Three tests were performed for each concentration from the calibration curve on different sensors, and the data are presented as the average values of results. The sensitivity of the sensor, expressed as the slope of the calibration graph, was calculated from the plot as 0.14 μA·μM^−1^ by linear regression. In addition, the limit of detection (LOD) was estimated at 66.90 nM based on S/N = 3.

### 3.6. Specificity, Intra- and Inter-Assay Precision, Reusability and Stability of the PyoV Sensor

Specificity tests were performed for the optimized sensor under the same conditions, and different common interfering substances were used (e.g., G, AA, ASA, DA, NADH, and UA). When using electrochemical methods on real samples (serum, urine, saliva), it is usually necessary to dilute with the electrolyte to enhance the conductivity. Depending on the dilution level (1:100) the chosen concentration of PyoV was found to be around 25 µM. Furthermore, the physiological concentration range of UA is 2201–5472 µM [18], and after a 1:100 dilution level the employed value could be included in the 22–54.72 µM range. AA has a physiological concentration range of 26–84.6 µM, close to the established concentration value of 25 µM [19]. The concentration of G was selected to be slightly higher that the physiological one situated between 4400 and 6100 µM (79.2–110 mg·mL^−1^) [20]. Pyocianin (PyoC), the other virulence factor for *P. aeruginosa*, was used as the negative control target to test the selectivity of the sensor for PyoV detection. As can be seen from Figure 6A, the current responses obtained with the elaborated sensor for 25 µM PyoV in the presence of 7000 μM G, 25 μM AA, 25 μM ASA, 25 μM NADH, 25 µM UA, and 25 μM DA presented reduced (or low) influences compared with the DPV signal registered for PyoV alone. Thus, the correspondent calculated recoveries between 96.51% and 115.46%. In the case of the PyoC standard solution prepared for the interference tests, an electrochemical signal was observed for PyoV also, as can be seen in Figure 6B (green curve (c)). This may have been due to the fact that both compounds are metabolites for *P. aeruginosa*, and could have been simultaneously present during the synthesis and separation of PyoC used for this study. Thus, when 25 µM PyoV and 10 µM PyoC were both in the solution, the intensity of the oxidation peak for the PyoV at about 0.3 V/Ag/AgCl increased, compared with the standard test of the same concentration (Figure 6B, blue (d), and Figure 6B, red (b), respectively), probably due to traces of PyoV from the PyoC solution.

The intra- and inter-assay precision of the sensor were investigated by testing the same concentration of PyoV (25 µM) performing five different tests on the same sensor and five different sensors, respectively. The DPV curves obtained during these tests were evaluated and the RSD was calculated. The average of the current intensity for the oxidation peak of PyoV was 3.269 µA, and the RSD of the intra-assay and inter-assay precisions of the sensor were 4.16% and 3.97%. 

The possibility to reuse the same sensor for more tests was also evaluated. Thus, different tests were performed using the same sensor in 25 µM of PyoV solution, after a washing step in PBS. It was observed that the current intensity slightly decreased after each test, the signal recovery being 97.25% after the second test, 91.94% after the fourth test, and 76.10% after the ninth test (Figure 7). We assume that each sensor could be reused with good analytical parameters for two successive tests, suitable for a disposable detection device.

To check the operational stability of this sensor, the electrochemical performance was tested against the storage time. Different PyoV sensors were elaborated using the optimized protocol and were stored in a dry and clean environment at room temperature (20 °C) if not in use. Three of these sensors were tested immediately after elaboration in 25 μM PyoV, while others were tested from time to time for 30 days. It was observed that the signal obtained for 25 μM PyoV increased by 8.75% after 3 days, 14.99% after 7 days, 10.44% after 14 days, while only by 4.15% after 30 days. These results suggest that the elaborated sensor exhibited good stability for the DPV determination of PyoV.

### 3.7. Recovery Tests of the PyoV Sensor in Real Samples

The sensor response towards 25 µM of PyoV spiked in different complex matrices such as human serum, saliva, and tap water was also tested. The spiked samples were prepared after the dilution of the real samples with electrolytes, and injected with the target analyte in order to achieve the desired concentrations. The results were expressed as recoveries, and it can be noticed that the values obtained for 25 µM PyoV ranged from 96.75% in saliva, to 98.83% in serum, and 100.55% in tap water (data not presented). The RSD of three different measurements of the same sample varied from 1.9% to 4.1%, respectively

## 4. Discussion

Until now bacterial infections are usually diagnosed by using microbiological techniques (cell culturing in controlled experimental conditions) in which case detection is based on cell growth and development. Clinical diagnosis is not possible if the concentration of bacteria in collected real samples does not exceed 10^5^ cells/mL [21]. In laboratories, the use of a simple method and an optimized platform for direct bacterial detection is not yet possible. A useful alternative for *P. aeruginosa* direct detection could be the use of PyoV, a virulence factor, as a marker with electrochemical activity.

A procedure based on the electrochemical oxidation of PyoV could allow the rapid and sensitive detection of *P. aeruginosa.* After two simple electrochemical conditioning steps, a commercially available SPE was successfully applied for the sensitive detection of PyoV even in the presence of several analytes known as possible interfering compounds in real samples. In this work, the ability to detect and quantify PyoV, as bacterial marker, in complex samples was demonstrated, offering a promising tool for early diagnosis of bacterial infections.

Electrochemical reduction of graphene was introduced as a conditioning step in the experimental protocol, in order to eliminate any small differences between electrodes. This step also allowed the reduction of the impurities that could possibly be found on the working electrode. Usually, this procedure is used for graphene oxide-modified electrodes in order to avoid the decrease in conductivity and carrier mobility observed when compared with graphene-modified electrodes [22,23]. The electrochemically reduced graphene has a larger number of electroactive sites, having a similar structure with that of graphene [24,25,26]. Moreover, the SEM images confirmed the generation of a more homogenous film after the electrochemical reduction of the graphene suitable for the detection of the target analyte.

The increase in peak current of 55% after the deposition of AuNPs could be assigned to the enhancement of the active surface by metallic NPs that facilitated the redox probe electrochemical transformation. Another explanation could be the good conductivity of AuNPs on the surface of the electrode, which were favorable for the electron transfer. The peak-to-peak separation became 0.227 V. The electroactive surface of the electrode was calculated after each modification step according to the Randles–Sevcik equation (Equation (1)) [27]:(1)$$I_{p}=2.69\cdot 10^{5}\cdot A\cdot D^{1/2}\cdot n^{3/2}\cdot v^{1/2}\cdot C$$
where I_p_ represents the value of the current peak intensity A; A is the effective area of the electrode surface (cm^2^); D is the diffusion coefficient of [Fe(CN)_6_]^3−/4−^ (6.70 × 10^−6^ cm^2^·s^−1^); *n* is the number of electrons involved in the redox reaction (1); *v* is the scan rate of the potential (V·s^−1^); and C is the bulk concentration of the redox probe (mol·cm^−3^).

By calculation, the effective surface area of the bare electrode was 0.150 cm^2^, while after decoration with AuNPs it became 0.234 cm^2^, showing that this step generated an important increase in the surface area of the working electrode.

Moreover, EIS was used as an effective and rapid technique to measure the electrical properties of the electrode before and after functionalization with different materials. The typical Nyquist representation includes a semicircle corresponding to the electron transfer limited process, the diameter displaying the electron transfer resistance values (R_ct_), and a linear zone that corresponds to the diffusion limited process.

When evaluating the analytical performance of the sensor, the low value of LOD could be attributed to the following factors: (i) reduced graphene with a high electron-transfer capability and large specific surface area, and (ii) excellent electrical conductivity of AuNPs determining the improvement of the electron transfer rate.

The detection of PyoV in the presence of G, AA, ASA, NADH, UA, and DA was not influenced, while in the presence of PyoC it could not be detected with high accuracy. Considering that the electrochemical oxidation of PyoC occurred at negative potential (−0.300 vs. Ag/V (pseudo Ag/AgCl)) it is unlikely that the presence of this compound would influence the signal of PyoV that appeared at a positive potential value (+0.325 vs. Ag/V (pseudo Ag/AgCl)). Indeed, from the DPV curve recorded in the presence of both compounds in the same sample (Figure 6B, blue curve (d)) it was observed that the signal for PyoV summed up the initial signal observed in the PyoC solution and the signal corresponding to the PyoV added concentration (25 μM). On the contrary, the electrochemical signal of PyoC was dramatically influenced by the presence of PyoV, proving that the two compounds cannot be determined simultaneously with this sensor. Hence, the sensor had a high specific affinity for PyoV detection, but cannot be used for the simultaneous detection of both *P. aeruginosa* virulence factors.

## 5. Conclusions

The ability of the DPV detection to sense PyoV in a micromolar range in complex matrices such as tap water, human serum, and saliva was demonstrated, being a practical validation for the sensor. Those satisfying results demonstrate that the sensor has a great potential for practical applications in the testing and quantification of PyoV in biological samples. 

The good sensitivity in detection was achieved through defining the optimal potential range for the PyoV voltammetric quantification with constant background contribution. The sensor was able to detect PyoV in a nanomolar range, proving the presence of *P. aeruginosa* in the analyzed samples. An important issue is the fact that the sensor can be reused a minimum of two times after cleaning with water, significantly reducing time and cost, qualifying it for fast diagnosis of *P. aeruginosa*, and making it suitable for point-of-care devices, as no sample pretreatment is needed. This novel electrochemical sensor successfully integrated the advantages of the high conductivity and large specific surface area of graphene, and the excellent electrochemical activity of AuNPs, with a high sensitivity, selectivity, wide linear range, and fast analysis time for PyoV.

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
