# Peer review of "A Nanocomposite Based on Reduced Graphene and Gold Nanoparticles for Highly Sensitive Electrochemical Detection of Pseudomonas aeruginosa through Its Virulence Factors"

_materials, 2019, doi:10.3390/ma12071180_

Round 1
Reviewer 1 Report
Lines 68-71: Authors have published two other manuscripts which are mentioned as [16,17]. Is the novelty of the presented study described in lines 71 onwards? Please make the new achievements clearer and highlight the differences with previous studies.
In scheme 1, the SEMs can be removed. They are unclear in this picture.
Figure1A: Please provide information about the reduction reaction which is shown in figure 1A.
Figure1A and Figure1B: The used signs to show the order of cycles are not clear.
Caption for figure 3 can be rewritten in a clearer manner.
Section 3.4.: From this section, I learned that preliminary reduction of graphene has an important effect on a further surface modification with AuNPs (homogenous and denser). Please rewrite this section to highlight that more.
In lines 226: cm2, which is used to shoe the sensitivity, is not available in x-axis of plot figure5B. Please make it clear.
The section specificity, which is an important evaluation step, is not clearly described. Especially figure6B needs more explanation. Why figure6B (a) has a signal? Why figure6B (b) shows about 35 µA for 25 µM PyoV, while figure3 (d) shows just 15 µA for 100 µM PyoV? Why potentials are shifted in figure 6B?
In section 3.7.: real samples were spiked with 25 µM. It is already mentioned in section 3.6. (Lines 235-237) that real samples are needed to be diluted with electrolyte 1:100. Did authors dilute the spiked samples?
Line 342: Please replace the used “ultra sensitivity” with a more suitable one.
Line 344: I think that proposed method could be used for determination of PyoV in µM range.
Author Response
Responses to Reviewer #1
We would like to thank to the reviewer for the useful suggestions which aim to improve the technical and scientific aspect of our work. All the suggested modifications performed in the manuscript are highlighted in yellow and are explained below.
Lines 68-71: Authors have published two other manuscripts which are mentioned as [16,17]. Is the novelty of the presented study described in lines 71 onwards? Please make the new achievements clearer and highlight the differences with previous studies.
The modification in order to highlight the protocol upgrade was rephrased in lines 74-78 and highlighted in yellow in the main text. The best analytical performance from the three synthesized platforms was obtained when the graphene and electrochemically generated AuNPs were associated. We observed that this synergic action was superior than the one generated on the chemical modified graphene. In order to highlight this feature, we inserted in the introduction the following phrase:
“Thus, the protocol was adapted to the electrochemical deposition of AuNPs on the graphene/graphite working electrode instead of using chemically Au modified graphene. The upgrade in the development protocol was closely linked to the increase of the stability of the platform as well as its sensitivity. Moreover, the polymeric film was eliminated, simplifying and reducing the time of the elaboration.”
In scheme 1, the SEMs can be removed. They are unclear in this picture.
The SEM images were removed according to the reviewers’ suggestion. In fact, the whole scheme was modified and all the unclear details were removed or replaced for better resolution.
Figure1A: Please provide information about the reduction reaction which is shown in figure 1A.
The electrochemical reduction of graphene was performed as a preconditioning step of the graphite/graphene working surface. This step allowed also the reduction of the impurities that could be possibly found on the working electrode. Moreover, the SEM images confirmed the generation of a more homogenous film after the electrochemical reduction of the graphene suitable for the detection of the target analyte.
Figure 1A and Figure1B: The used signs to show the order of cycles are not clear.
The following text was inserted in the Results section in order to clear the signification of the arrows inserted in the CV graphs:
As it can be observed in Figure 1(A), the signal decreases proportionally with the number of cycles. The electrochemical deposition of AuNPs was confirmed by the increase of both anodic and cathodic peaks corresponding to the oxidation/reduction of Au on the graphene/graphite surface, as it can be seen from Figure 1(B).
Caption for figure 3 can be rewritten in a clearer manner.
The caption for Figure 3 was rewritten and highlighted in yellow in the main text:
Figure 3 DPVs performed in PBS in the absence of PyoV at reduced graphene/graphite SPE modified with AuNPs (black, a) and performed in 100 µM PyoV in PBS at: bare graphene/graphite SPE (red, b); reduced graphene/graphite SPE (green, c) and reduced graphene/graphite SPE with AuNPs (blue, d) (20 mM PBS pH 7.4; from −0.3 to +0.8 V vs. Ag/AgCl; 0.02 Vs-1)
Section 3.4.: From this section, I learned that preliminary reduction of graphene has an important effect on a further surface modification with AuNPs (homogenous and denser). Please rewrite this section to highlight that more.
The following text was inserted in the main text, section 3.4 and highlighted in yellow.
As it can be also seen, the coverage with AuNPs was higher after the conditioning step confirming furthermore the importance of the conditioning protocol regarding the density in catalytic metallic NPs.
In lines 226: cm2, which is used to shoe the sensitivity, is not available in x-axis of plot figure5B. Please make it clear.
We thank the reviewer for this important observation! Actually, the unit cm2 was introduced accidentally. This is necessary when current density is considered on Y axis of the calibration plot. We have considered the current, thus the right unit in our study is in fact µA µM-1. We have performed the right corrections in text and the modifications are highlighted.
The section specificity, which is an important evaluation step, is not clearly described. Especially figure6B needs more explanation. Why figure6B (a) has a signal? Why figure6B (b) shows about 35 µA for 25 µM PyoV, while figure3 (d) shows just 15 µA for 100 µM PyoV? Why potentials are shifted in figure 6B?
In Figure 6B the DPV curves were presented on larger potential scale compared with the other ones in the manuscript in order to contain both the oxidation signal of pyoverdine (at about 0.3 V) and pyocyanine (at about -0.2 V). Thus, the 35 µA signal mentioned by the reviewer represents in fact the baseline of the DPVs, as it can be observed in the absence of the analytes in solutions (Figure 6(B) (black, a). The paragraph was modified and more detailed discussions were introduced for Figure 6(B), as it can be seen below:
In the case of PyoC standard solution prepared for the interference tests, an electrochemical signal was observed also for PyoV, as can be seen in Figure 6(B) (green curve, c)). This may be due to the fact that both compounds are metabolites for P. aeruginosa and could be simultaneously present during the synthesis and separation of PyoC used for this study. Thus, when 25 μM PyoV and 10 μM PyoC are both in solution, the intensity of the oxidation peak for the PyoV at about 0.3 V/Ag/AgCl is increasing compared to the standard test of the same concentration (Figure 6(B) (blue, d) and Figure 6(B) (red, b), respectively), probably due to traces of PyoV from the PyoC solution.
In section 3.7.: real samples were spiked with 25 µM. It is already mentioned in section 3.6. (Lines 235-237) that real samples are needed to be diluted with electrolyte 1:100. Did authors dilute the spiked samples?
The spiked sample were prepared after the dilution of the real samples with electrolyte and injected with the target analyte in order to achieve the desired concentrations. The PyoV samples used for the calibration curve were prepared directly in 20 mM PBS pH 7.4.
Line 342: Please replace the used “ultra sensitivity” with a more suitable one.
The ultra sensitivity was replaced with good sensitivity and highlighted in yellow in the main text.
Line 344: I think that proposed method could be used for determination of PyoV in µM range.
The range of 0.5 μM–100 μM“ was replaced by “micromolar range“ and highlighted in yellow in the main text.

Reviewer 2 Report
Please see the attached file

Author Response
Responses to Reviewer #2
We would like to thank to the reviewer for recognizing the quality of our study and for the useful suggestions which aim to improve the technical and scientific aspect of our work. All the suggested modifications performed in the manuscript are highlighted in yellow and are explained below.
Reviewer 2. In this paper the authors presented an electrochemical sensor based on reduced graphene and gold nanoparticles as a tool for an early diagnosis of infections by Pseudomonas aeruginosa. The paper is original and interesting and is written in good English. The title clearly describes the article, and the abstract reflects the content and the goal of the article. The introduction was exhaustive with a precise use of related literature. The authors have clearly in mind the goal of the paper as well as the approach to reach it. Nevertheless, in this paragraph, the goal was reported (line73-78) as an obtained result and not as an objective. This should be modified. The analytical approach was good, the experimental section is complete with an accurate description of the system, of the preparation steps and of the working principle. Results are clearly reported and followed by an exhaustive discussion and interesting conclusions.
Minor corrections are requested:
- Line 33-35: Normally, if the name of a bacterial species is reported in full at a beginning of a manuscript (i.e. Pseudomonas aeruginosa), the short form (P. aeruginosa) can be used later and it is not necessary to report it in brackets. Please modify;
The modification was performed and the short term was used as indicated by the reviewer.
- Scheme 1: X and Y axis of the inset (voltammogram) are not readable even on a large screen: please provide a graph with a better definition;
The X/Y axes from the inset of the voltammogram were enhanced in order to provide a better resolution. The SEM images were removed according to the reviewers’ suggestion. In fact, the whole scheme was modified and all the unclear details were removed or replaced for better resolution.
- Scheme 1: please provide a more exhaustive and detailed caption;
Schematic representation of the PyoV sensor elaboration protocol and its testing principle. Was replaced by the following text:
Schematic representation of the PyoV sensor elaboration protocol ((1) the electrochemical reduction of the graphene by CV, followed by (2) the electrochemical deposition of the AuNPs by CV) and testing protocol by DPV in the presence of PyoV, as target analyte.
Figure 5 (A): please provide a correspondence between color lines and concentrations in the figure caption;
The colours of the lines were inserted and highlighted in yellow in the main text, as follows:
DPVs in 20 mM PBS pH 7.4 without PyoV (black line) and for different concentrations of PyoV: 0.5 (green line); 0.75 (red line); 1 (dark blue line); 5 (light blue line); 10 (purple line); 25 (yellow line); 50 (kaki line); 75 (blue line) and 100 (dark purple line)
- Line 259-260: the sentence “… which can be considered promising for further applications” was not supported by any reference, so it can be considered an authors’ conclusion. Please provide bibliography or move the sentence to the “Discussion” or “Conclusions” paragraph.
The sentence “respectively, which can be considered promising for further applications“ was deleted. A similar formulation was already stated in the Conclusions paragraph as:
“Those satisfying results demonstrated that the sensor had a great potential for practical application for testing and quantification of PyoV in biological samples“.
It is reviewer’s opinion that the manuscript is original and scientifically interesting and should be accepted after the above-mentioned minor corrections.

Round 2
Reviewer 1 Report
Please insert the authors’ answer: “The electrochemical reduction of graphene was performed as a preconditioning step of the graphite/graphene working surface. This step allowed also the reduction of the impurities that could be possibly found on the working electrode. Moreover, the SEM images confirmed the generation of a more homogenous film after the electrochemical reduction of the graphene suitable for the detection of the target analyte.” in a suitable position in manuscript.
Please provide this information: “The spiked sample were prepared after the dilution of the real samples with electrolyte and injected with the target analyte in order to achieve the desired concentrations.” in a suitable position in manuscript.
Author Response
Answer for Reviewer#2
We thank the reviewer for the useful suggestions. All the modifications were highlighted in yellow in the manuscript.
Please insert the authors’ answer: “The electrochemical reduction of graphene was performed as a preconditioning step of the graphite/graphene working surface. This step allowed also the reduction of the impurities that could be possibly found on the working electrode. Moreover, the SEM images confirmed the generation of a more homogenous film after the electrochemical reduction of the graphene suitable for the detection of the target analyte.” in a suitable position in manuscript.
The suggested sentence was placed in Discussion chapter.
Please provide this information: “The spiked sample were prepared after the dilution of the real samples with electrolyte and injected with the target analyte in order to achieve the desired concentrations.” in a suitable position in manuscript.
The suggested sentence was placed in Discussion chapter.
